# Reward Constrained Interactive Recommendation with Natural Language Feedback

**Ruiyi Zhang**[1*]**, Tong Yu**[2*] **, Yilin Shen**[2]**, Hongxia Jin**[2]**, Changyou Chen**[3]**, Lawrence Carin**[1]
[1] Duke University, [2] Samsung Research America, [3] University at Buffalo

## Abstract

Text-based interactive recommendation provides richer user feedback and has demonstrated advantages over traditional interactive recommender systems. However, recommendations can easily violate preferences of users from their past natural-language feedback, since the recommender needs to explore new items for further improvement. To alleviate this issue, we propose a novel constraint-augmented reinforcement learning (RL) framework to efficiently incorporate user preferences over time. Specifically, we leverage a discriminator to detect recommendations violating user historical preference, which is incorporated into the standard RL objective of maximizing expected cumulative future rewards. Our proposed framework is general and is further extended to the task of constrained text generation. Empirical results show that the proposed method yields consistent improvement relative to standard RL methods.

## 1 Introduction

Traditional recommender systems depend heavily on user history. However, these approaches, when implemented in an offline manner, cannot provide satisfactory performance due to sparse history data and unseen dynamic new items (*e.g.*, new products, recent movies, etc.). Recent work on recommender systems has sought to interact with users, to adapt to user preferences over time. Most existing interactive recommender systems are designed based on simple user feedback, such as clicking data or updated ratings [6, 29, 32]. However, this type of feedback contains little information to reflect complex user attitude towards various aspects of an item. For example, a user may like the graphic of a dress but not its color. A click or numeric rating is typically not sufficient to express such a preference, and thus it may lead to poor recommendations. By contrast, allowing a recommender system to use natural-language feedback provides richer information for future recommendation, especially for visual item recommendation [19, 20]. With natural-language feedback, a user can describe features of desired items that are lacking in the current recommended items. The system can then incorporate feedback and subsequently recommend more suitable items. This type of recommendation is referred to as *text-based interactive recommendation*.

Flexible feedback with natural language may still induce undesired issues. For example, a system may ignore the previous interactions and keep recommending similar items, for which a user has expressed the preference before. To tackle these issues, we propose a reward constrained recommendation (RCR) framework, where one sequentially incorporates constraints from previous feedback into the recommendation. Specifically, we formulate the text-based interactive recommendation as a constraint-augmented reinforcement learning (RL) problem. Compared to standard constraint-augmented RL, there are no explicit constraints in text-based interactive recommendation. To this end, we use a discriminator to detect violations of user preferences in an adversarial manner. To further validate our proposed RCR framework, we extend it to constrained text generation to discourage undesired text generation.

* Equal contribution. Work done while RZ was a part-time research intern at Samsung Research America.

The main contributions of this paper are summarized as follows. (*i*) A novel reward constrained recommendation framework is developed for text-based interactive recommendation, where constraints work as a dynamically updated critic to penalize the recommender. (*ii*) A novel way of defining constraints is proposed, in an adversarial manner, with better generalization. (*iii*) Extensive empirical evaluations are performed on text-based interactive recommendation and constrained text generation tasks, demonstrating consistent performance improvement over existing approaches.

## 2 Background

### 2.1 Reinforcement Learning

Reinforcement learning aims to learn an optimal policy for an agent interacting with an unknown (and often highly complex) environment. A policy is modeled as a conditional distribution $\pi(\boldsymbol{a}|\boldsymbol{s})$, specifying the probability of choosing action $\boldsymbol{a} \in \mathcal{A}$ when in state $\boldsymbol{s} \in \mathcal{S}$. Formally, an RL problem is characterized by a Markov decision process (MDP) [38], $\mathcal{M} = \langle \mathcal{S}, \mathcal{A}, P, R \rangle$. In this work, we consider recommendation for finite-horizon environments with the average reward criterion. If the agent chooses action $\boldsymbol{a} \in \mathcal{A}$ at state $\boldsymbol{s} \in \mathcal{S}$, then the agent will receive an immediate reward $r(\boldsymbol{s}, \boldsymbol{a})$, and the state will transit to $\boldsymbol{s}' \in \mathcal{S}$ with probability $P(\boldsymbol{s}'|\boldsymbol{s}, \boldsymbol{a})$. The expected total reward of a policy $\pi$ is defined as [42]:

$$J_R(\pi) = \sum_{t=1}^{\infty} \mathbb{E}_{P,\pi}\left[r(\boldsymbol{s}_t, \boldsymbol{a}_t)\right]. \tag{1}$$

In (1) the sum is over infinite time steps, but in practice we will be interested in finite horizons.

The goal of an agent is to learn an optimal policy that maximizes $J_R(\pi)$. A constrained Markov decision process (CMDP) [3] extends the MDP framework by introducing the constraint $C(\boldsymbol{s}, \boldsymbol{a})$ (mapping a state-action pair to costs, similar to the usual reward) [2] and a threshold $\alpha \in [0, 1]$. Denoting the expectation over the constraint $C(\boldsymbol{s}, \boldsymbol{a})$ as $J_C(\pi) = \sum_{t=1}^{\infty} \mathbb{E}_{P,\pi}[C(\boldsymbol{s}_t, \boldsymbol{a}_t)]$, the constrained policy optimization thus becomes [1]:

$$\max_{\pi \in \Pi} J_R(\pi), \quad \text{s.t. } J_C(\pi) \le \alpha . \tag{2}$$

### 2.2 Text-based Interactive Recommendation as Reinforcement Learning

We employ an RL-based formulation for sequential recommendation of items to users, utilizing user feedback in natural language. Denote $\boldsymbol{s}_t \in \mathcal{S}$ as the state of the recommendation environment at time $t$ and $\boldsymbol{a}_t \in \mathcal{A}$ as the recommender-defined items from the candidate items set $\mathcal{A}$. In the context of a recommendation system, as discussed further below, the state $\boldsymbol{s}_t$ corresponds to the state of sequential recommender, implemented via a LSTM [23] state tracker. At time $t$, the system recommends item $\boldsymbol{a}_t$ based on the current state $\boldsymbol{s}_t$ at time $t$. After viewing item $\boldsymbol{a}_t$, a user may comment on the recommendation in natural language (a sequence of natural-language text) $\boldsymbol{x}_t$, as feedback. The recommender then receives a reward $r_t$ and perceives the new state $\boldsymbol{s}_{t+1}$. Accordingly, we can model the recommendation-feedback loop as an MDP $\mathcal{M} = \langle \mathcal{S}, \mathcal{A}, P, R \rangle$, where $P : \mathcal{S} \times \mathcal{A} \times \mathcal{S} \mapsto \mathbb{R}$ is the environment dynamic of recommendation and $R : \mathcal{S} \times \mathcal{A} \mapsto \mathbb{R}$ is the reward function used to evaluate recommended items. The recommender seeks to learn a policy parameterized by $\boldsymbol{\theta}$, *i.e.*, $\pi_{\boldsymbol{\theta}}(\boldsymbol{a}|\boldsymbol{s})$, that corresponds to the distribution of items conditioned on the current state of the recommender. The recommender is represented as an optimal policy that maximizes the expected reward as $J_R(\pi) = \sum_t \mathbb{E}_{P,\pi}\left[r(\boldsymbol{s}_t, \boldsymbol{a}_t)\right]$. At each time step, the recommender sequentially selects potential desired items at each time step via $\boldsymbol{a}_t = \arg\max_{\boldsymbol{a} \in \mathcal{A}} \pi_{\boldsymbol{\theta}}(\boldsymbol{a}|\boldsymbol{s}_t)$.

## 3 Proposed Method

In text-based interactive recommendation, users provide natural-language-based feedback. We consider the recommendation of visual items [19, 20]. As shown in Figure 2, the system recommends an item to the user, with its visual appearance. The user then views the recommended item and gives feedback in natural language, describing the desired aspects that the current recommended item lacks. The system then incorporates the user feedback and recommends (ideally) more-suitable items, until the desired item is found. While users provide natural-language feedback on recommendations, standard RL methods may overlook the information from the feedback and recommend items that

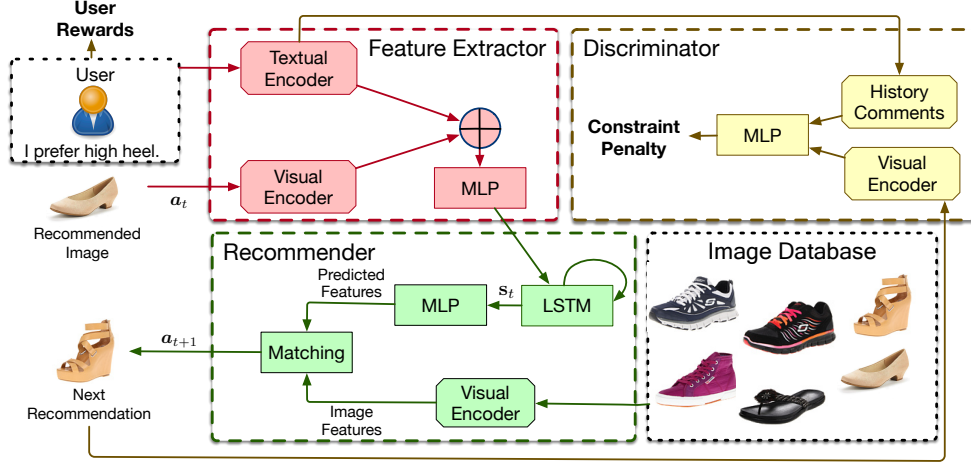

Figure 1: Overview of the reward constrained recommender model. When receiving the recommended images, the user gives natural-language feedback, and this feedback will be used for the next item recommendation, as well as preventing future violations.

violate the user's previous feedback. To better understand this issue, consider the example in Figure 2. In round 3, the system forgets, and recommends an item that violates previous user preference on the 'ankle boots'.

To alleviate this issue, we consider using feedback from users as constraints, and formulate text-based interactive recommendation as a constrained policy optimization problem. The

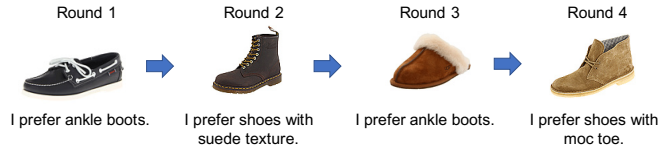

Figure 2: An example of text-based interactive recommendation.

difference between the investigated problem and conventional constrained policy optimization [3, 5] is that constraints are added sequentially, affecting the search space of a policy in a different manner. Our model is illustrated in Figure 1.

## 3.1 Recommendation as Constrained Policy Optimization

We consider an RL environment with a large number of discrete actions, deterministic transitions, and deterministic terminal returns. Suppose we have the user preference as constraints $J_C(\pi_\theta)$ when making recommendations. The objective of learning a recommender is defined as:

$$J_R(\pi_\theta) = \sum_{t=1}^{\infty} \mathbb{E}_{P,\pi_\theta}\left[r(\boldsymbol{s}_t, \boldsymbol{a}_t)\right], \ \text{s.t.} \ J_C(\pi_\theta) \le \alpha \ . \tag{3}$$

If one naively augments previous user preferences as a hard constraint, *i.e.*, exactly attributes matching, it usually leads to a sub-optimal solution. To alleviate this issue, we propose to use a learned constraint function based on the visual and textual information.

**Constraint Functions** In text-based interactive recommendation, we explicitly use the user preference as constraints. Specifically, we exploit user feedback and put it as sequentially added constraints. To generalize well on the constraints, we learn a discriminator $C_\phi$ parameterized by $\phi$ as the constraint function. We define two distributions on feedback-recommendation pairs, *i.e.*, non-violation distribution $p_r$, and violation distribution $p_f$ (details provided in Appendix A.2). The objective of the discriminator is to minimize the following objective:

$$L(\phi) = -\mathbb{E}_{(\boldsymbol{s},\boldsymbol{a})\sim p_f}\left[\log(C_\phi(\boldsymbol{s},\boldsymbol{a}))\right] - \mathbb{E}_{(\boldsymbol{s},\boldsymbol{a})\sim p_r}\left[\log(1 - C_\phi(\boldsymbol{s},\boldsymbol{a}))\right] . \tag{4}$$

With the discriminator as the constraint *i.e.*, $J_{C_\phi}(\pi_\theta) \triangleq \sum_{t=1}^{\infty} \mathbb{E}_{P,\pi_\theta}[C_\phi(\boldsymbol{s}_t, \boldsymbol{a}_t)]$, the constraint is imposed. However, directly solving the constrained-optimization problem in (3) is difficult, and we employ the Lagrange relaxation technique [4] to transform the original objective to an equivalent problem as:

$$\min_{\lambda \ge 0} \max_{\theta} L(\lambda, \theta, \phi) = \min_{\lambda \ge 0} \max_{\theta} \left[J_R(\pi_\theta) - \lambda \cdot (J_{C_\phi}(\pi_\theta) - \alpha)\right] \ , \tag{5}$$

where $\lambda \geq 0$ is a Lagrange multiplier. Note that as $\lambda$ increases, the solution to (5) converges to that of (3). The goal is to find a saddle point $(\theta^*(\lambda^*), \lambda^*)$ of (5), that can be achieved approximately by alternating gradient descent/ascent. Specifically, the gradient of (5) can be estimated using policy gradient [42] as:

$$\nabla_\theta L(\theta, \lambda, \phi) = \mathbb{E}_{P,\pi}\left[\left(r(\boldsymbol{s}_t, \boldsymbol{a}_t) - \lambda C_\phi(\boldsymbol{s}_t, \boldsymbol{a}_t)\right) \nabla_\theta \log \pi_\theta(\boldsymbol{s}_t, \boldsymbol{a}_t)\right], \quad (6)$$

$$\nabla_\lambda L(\theta, \lambda, \phi) = -\left(\mathbb{E}_{P,\pi}[C_\phi(\boldsymbol{s}_t, \boldsymbol{a}_t)] - \alpha\right), \quad (7)$$

where $C_\phi(\boldsymbol{s}_t, \boldsymbol{a}_t)$ is the general constraint, specified in the following.

**Penalized Reward Functions**    Note that the update in (6) is similar to the actor-critic method [42]. While the original use of a critic in reinforcement learning was for variance reduction [42], here we use it to penalize the policy for constraint violations. In order to ensure the constraints, $\lambda$ is also optimized using policy gradient via (7). The optimization proceeds intuitively as: *i*) when a violation happens (*i.e.*, $C_\phi(\boldsymbol{s}, \boldsymbol{a}) > \alpha$), $\lambda$ will increase to penalize the policy. *ii*) If there is no violation (*i.e.*, $C_\phi(\boldsymbol{s}, \boldsymbol{a}) < \alpha$), $\lambda$ will decrease to give the policy more reward.

**Model Training**    We alternatively update the constraint function, *i.e.*, the discriminator and the recommender $\pi_\theta$, similar to the Generative Adversarial Network (GAN) [15]. Specifically, the parameters are updated via the following rules:

$$\theta_{k+1} = \Gamma_\theta[\theta_k + \eta_1(k)\nabla_\theta L(\lambda_k, \theta_k, \phi_k)], \quad (8)$$

$$\phi_{k+1} = \phi_k + \eta_2(k)\nabla_\phi L(\lambda_k, \theta_k, \phi_k) \ , \quad (9)$$

$$\lambda_{k+1} = \Gamma_\lambda[\lambda_k - \eta_3(k)\nabla_\lambda L(\lambda_k, \theta_k, \phi_k)] \ , \quad (10)$$

where $\Gamma_\theta$ is a projection operator, which keeps the stability as the parameters are updated within a trust region; $\Gamma_\lambda$ projects $\lambda$ into the range $[0, \lambda_{\max}]$.

We denote a three-timescale Reward Constrained Recommendation process, *i.e.*, the three parts are updated with different frequency and step sizes: the recommender aims to maximize the expected reward with less violations following (8). As described in the Algorithm 1, the discriminator is updated following (9) to detect new violations, and $\lambda$ is updated following (10).

---

**Algorithm 1** Reward Constrained Recommendation

**Input:** constraint $C(\cdot)$, threshold $\alpha$, learning rates $\eta_1(k) > \eta_2(k) > \eta_3(k)$
Initialize recommender and discriminator parameters with pretrained ones, Lagrange multipliers $\lambda_0 = 0$
**repeat**
    **for** $t = 0, 1, ..., T - 1$ **do**
        Sample action $a_t \sim \pi$, observe next state $s_{t+1}$, reward $r_t$ and penalties $c_t$
        $\hat{R}_t = r_t - \lambda_k c_t$
        **Recommender update** with (8)
    **end for**
    **Discriminator update** with (9)
    **Lagrange multiplier update** with (10)
**until** Model converges
**return** recommender (policy) parameters $\theta$

---

### 3.2   Model Details

We discuss details on model design when applying the proposed framework in a text-based recommender system.

**Feature Extractor**    Our feature extractor consists of the encoders of text and visual inputs. Similar to [20], we consider the case where the visual attributes are available. We encode the raw images of the items by ResNet50 [21] and an attribute network, *i.e.*, the visual feature $\boldsymbol{c}_t^{vis}$ of the item $\boldsymbol{a}_t$ is the concatenation of $\texttt{ResNet}(\boldsymbol{a}_t)$ and $\texttt{AttrNet}(\boldsymbol{a}_t)$. The input of the attribute network is an item's encoding by ResNet50 and the attribute network outputs this items' attribute values. We further encode the user comments in texts by an embedding layer, a LSTM and a linear mapping. Given a user comment $\boldsymbol{x}_t$, the final output of textual context is denoted as $\boldsymbol{c}_t^{txt}$. The encoded image and comment are further concatenated as the input to an MLP, and then the recommender component.

**Recommender**    With the visual feature $\boldsymbol{c}_t^{vis}$ and textual feature $\boldsymbol{c}_t^{txt}$, the recommender perceives the state in an auto-regressive manner. At time $t$, the state is $\boldsymbol{s}_t = f(g([\boldsymbol{c}_t^{vis}, \boldsymbol{c}_t^{txt}]), \boldsymbol{s}_{t-1})$, where $g$ is an MLP for textual and visual matching, and $f$ is the LSTM unit [23]. Since our goal in each user session is to find items with a set of desired attribute values, we use the policy $\pi_\theta$ with multi-discrete action spaces [22, 12]. For each attribute, the desired attribute value by the user is sampled from a categorical distribution. Given the state $\boldsymbol{s}_t$, the probability of choosing a particular attribute value is output by a three-layer fully connected neural network with a softmax activation function. The recommender samples the values of different attributes from $\pi_\theta$. If $K$ items are recommended at each time, we select the items that are top $K$ closest to the sampled attribute values under Euclidean distance in the visual attribute space.

**Discriminator** The discriminator is designed to discriminate whether a recommended item at time $t$ violates previous user comments in the current session. That is, given the visual feature of current image $\boldsymbol{c}_t^{vis}$, and textual features $\{\boldsymbol{c}_j^{txt}\}_{j=1}^{t-1}$, the discriminator outputs whether the image violates the user comment. In practice, this discriminator is a three-layer fully connected neural network and trained on-the-fly to incrementally learn the multimodal matching between the user comments and item visual features. Following Algorithm 1, we update the discriminator after each user session, where a user interacts with the system for several time steps, or quits. To further enhance the results, when making recommendations, we reject some items based on this discriminator. If an item $\boldsymbol{a}_t$ sampled by the recommender has high probability of violating the previous comments $\{\boldsymbol{x}_i\}_{i=1}^{t-1}$, we ignore this item and sample another item to recommend.

## 3.3 Extension to Constrained Text Generation

In this section, we describe how to extend our framework for constrained text generation.

We consider text generation with specific constraints. Specifically, we consider the scenario of controlling for negative sentiments. For example, a generator may generate some offensive or negative words, which will affect the user experience in some situations, such as with an online chatbot for helping consumers. To alleviate this issue, we applied the proposed RCR methods for text generation.

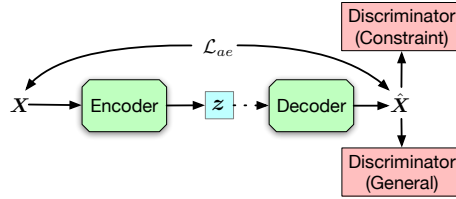

Figure 3: Overview of the constrained text-generation model: $\mathcal{L}_{ae}$ is the reconstruction term from the VAE in pretraining. The constraint will give a penalty when generated text violates the constraint discriminator.

We assume each sentence is generated from a latent vector $\boldsymbol{z} \sim p(\boldsymbol{z})$, where $p(\boldsymbol{z})$ is the distribution of a latent code. Text generation is then formulated as the learning of a distribution: $p(\boldsymbol{X}) = \int_{\boldsymbol{z}_x} p(\boldsymbol{X}|\boldsymbol{z}_x)q(\boldsymbol{z}_x|\boldsymbol{X})d\boldsymbol{z}_x$, where $p$ corresponds to a decoder and $q$ to an encoder model, within the encoder-decoder framework; $\boldsymbol{z}$ is the latent code containing content information. The generator learns a policy $\pi_\theta$ to generate a sequence $Y = (y_1, \ldots, y_T)$ of length $T$. Here each $y_t$ is a token from vocabulary $\mathcal{A}$. The objective is to maximize the expected reward with less constraint violations, defined as:

$$\mathcal{L}(\theta, \lambda, \phi) = \min_{\lambda \geq 0} \max_\theta \mathbb{E}_{Y \sim \pi_\theta} \left[ r(Y) - \lambda(C_\phi(Y) - \alpha) \right], \tag{11}$$

where $r$ is the reward function, that can be a metric reward (*e.g.*, BLEU) or a learned reward function with general discriminator [52]; $C_\phi(\cdot)$ is the constraint discriminator for the generation. In practice, we pretrain our generator $\pi_\theta$ with a variational autoencoder (VAE) [25], and we only use the decoder as our generator. More details about the pretrained model are provided in Appendix A.1. There is a constraint for the generation, and the framework is illustrated in Figure 3. The general discriminator can be a language model [48], and the constraint is a learned function parameterized by a neural network. During inference, the model generates text based on draws from an isotropic Gaussian distribution, *i.e.*, $\boldsymbol{z} \sim \mathcal{N}(\boldsymbol{0}, \boldsymbol{I})$. Here we only consider the static constraint with non-zero final deterministic reward.

## 4 Related Work

**Constrained Policy Optimization** Constrained Markov Decision Processes [3] are employed in a wide range of applications, including analysis of electric grids [26] and in robotics [8, 17]. Lagrange multipliers are widely used to solve the CMDP problem [43, 5], as adopted in our proposed framework. Other solutions of CMDP include use of a trust region [1], and integrating prior knowledge [11]. Additionally, some previous work manually selects the penalty coefficient [13, 31, 37]. In contrast with standard methods, our constraint functions are: (*i*) sequentially added via natural-language feedback; (*ii*) parameterized by a dynamically updated neural network with better generalization.

**Text-Based Recommender System** Communications between a user and recommendation system have been leveraged to understand user preference and provide recommendations. Entropy-based methods and bandits have been studied in question selection [34, 10]. Deep learning and reinforcement learning models have been proposed to understand user conversations and make recommendations [2, 9, 16, 41, 30, 53, 56]. Similar to [10, 41, 30, 53], the items are associated with a set of attributes in our recommendation setting. In the existing works, the content of the conversation serves as the

constraint when a system makes recommendations. However, in most existing works, constraints from the conversations are not explicitly modeled. By contrast, this paper proposes a novel constrained reinforcement learning framework to emphasize the constraints when making recommendations.

**Interactive Image Retrieval** Leveraging user feedback on images to improve image retrieval has been studied extensively [45]. Depending on the feedback format, previous works can be categorized into relevance feedback [39, 47] and relative-attributes feedback [27, 36, 49]. In these works, the attributes to describe the images are pre-defined and fixed. To achieve more flexible and precise representation of the image attributes, Guo, *et al.* [19] proposes an end-to-end approach, without pre-defining a set of attributes. Their goal is to improve the ranking of the target item, while we focus on recommending items that do not violate the users' previous comments in the iterative recommendation. Thus, we develop a different evaluation simulator as detailed in Section 5.1. In [53], it is assumed that an accurate discriminator pretrained on huge-amount offline data is available at the beginning, which is usually impractical. Instead, our novel RCR framework learns the discriminator from scratch and dynamically updates the model $\phi$ and its weight $\lambda$ by (9) and (10) online.

**Constrained Text Generation** Adversarial text generation [52, 7, 33, 14, 54, 35] use reinforcement learning (RL) algorithms for text generation. They use the REINFORCE algorithm to provide an unbiased gradient estimator for the generator, and apply the roll-out policy to obtain the reward from the discriminator. LeakGAN [18] adopts a hierarchical RL framework to improve text generation. GSGAN [28] and TextGAN [55, 24] use the Gumbel-softmax and soft-argmax representation, respectively, to deal with discrete data. Wang, *et al.* [46] put topic-aware priors on the latent codes to generate text on specific topics. All these works consider generating sentences with better quality and diversity, without explicit constraints.

# 5 Experiments

We apply the proposed methods in two applications: text-based interactive recommendation and constrained text generation, to demonstrate the effectiveness of our proposed RCR framework.

## 5.1 Text-Based Interactive Recommendation

**Dataset and Setup** Our approaches are evaluated on the UT-Zappos50K dataset [50, 51]. UT-Zappos50K is a shoe dataset consisting of $50,025$ shoe images. This dataset provides rich attribute data and we focus on shoes category, shoes subcategory, heel height, closure, gender and toe style in our evaluation. Among all the images, $40,020$ images are randomly sampled as training data and the rest are used as test data. To validate the generalization ability of our approach, we compare the performance on seen items and unseen items. The seen items are the items in the training data where the item visual attributes are carefully labeled. The unseen items are the items in the test data. We assume the unseen items are newly collected and have no labeled visual attributes. We train the attribute network on the training data, under the cross-entropy loss. The ResNet50 is pretrained on ImageNet and is fixed subsequently. When we report the results on seen and unseen items, their attribute values are predicted by the attribute network. We pretrain the textual encoder, where the labels are the described attribute values, under the cross-entropy loss. The training data consists of the comments collected by annotators as detailed later in this section. In reinforcement learning, we use Adam [25] as the optimizer. We set $\alpha = 0.5$ and $\lambda_{\max} = 1$.

We define the reward as the visual similarity between the recommended and desired items. Similar to [20], in our task both images and their visual attributes are available to measure the similarity. It is desired that the recommended item becomes more similar to the desired item with more user interactions. Thus, at time $t$, given the recommended item $\boldsymbol{a}_t$ and the desired item $\boldsymbol{a}^*$, we want to minimize their visual difference. In detail, we maximize the following visual reward $r_t = -||\texttt{ResNet}(\boldsymbol{a}_t) - \texttt{ResNet}(\boldsymbol{a}^*)||_2 - \lambda_{att}||\texttt{AttrNet}(\boldsymbol{a}_t) - \texttt{AttrNet}(\boldsymbol{a}^*)||_0$, where $||\cdot||_2$ is the $\mathcal{L}_2$ norm, $||\cdot||_0$ is the $\mathcal{L}_0$ norm, and we set $\lambda_{att} = 0.5$ to ensure the scales of the two distances are similar. If the system is not able to find the desired item before $50$ interactions, we will terminate this user session and the system will receive an extra reward $-3$ (*i.e.*, a penalty).

**Online Evaluation** We cannot directly detect the violations with existing text-based interactive recommendation dataset [19], since there are no attribute labels for the images. A recent relevant fashion dataset provides the attribute labels [3] derived from the text metadata [20]. Unfortunately, we

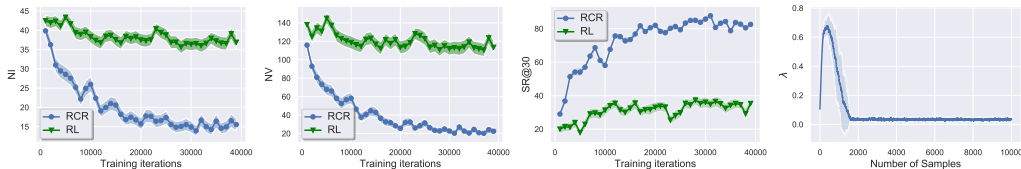

Figure 4: Number of Interactions (NI), Number of Violations (NV), Success Rate@30 (SR@30) with respect to training iterations and the values of $\lambda$ in RCR with respect to number of samples. The RL method converges much slower than the RCR.

|  | **SR@10 ↑** | **SR@20 ↑** | **SR@30 ↑** | **NI ↓** | **NV ↓** |
|---|---|---|---|---|---|
| RL (Unseen) | 19% | 44% | 63% | $26.75 \pm 1.67$ | $70.02 \pm 6.20$ |
| RL + Naive (Unseen) | 52% | 83% | 94% | $12.72 \pm 0.93$ | $16.47 \pm 2.75$ |
| RCR (Unseen) | 74% | 86% | 94% | $10.91 \pm 1.06$ | $11.32 \pm 1.98$ |
| RCR (Seen) | 78% | 91% | 92% | $10.34 \pm 1.18$ | $12.25 \pm 2.99$ |

Table 1: Comparisons between different approaches. Except the row of RCR (seen) reporting results on training data, all the results are on the test data with unseen items.

observe that the user's comments are usually unrelated to the attribute labels. Therefore, we need to collect the user's comments relevant to attributes with groundtruth, for our evaluation purpose.

Further, evaluating the proposed system requires the ability to get access to all user reactions to any possible items at each time step. For the evaluation on the UT-Zappos50K dataset, we use a similar *simulator* to Guo, *et al.* [19]. This simulator acts as a surrogate for real human users by generating their comments in natural language. The generated comments describe the prominent visual attribute differences between any pair of desired and candidate items.

To achieve this, we collect user comments relevant to the attributes with groundtruth and train a user simulator. A training dataset is collected for 10,000 pairs of images with visual attributes. These pairs are prepared such that in each pair there is a recommended item and a desired item. Given a pair of images, one user comment is collected. The data are collected in a scenario in which the customer talks with the shopping assistant to get the desired items. The annotators act as the customers to express the desired attribute values of items. For the evaluation purpose, we adopt a simplified setting and instruct the annotators to describe the comments related to a fixed set of visual attributes. Thus, the comments in our evaluation are relatively simpler compared to the real-world sentences. Considering this, we further augment the collected user comment data as follows. From the real-world sentences collected from annotators, we derive several sentence templates. Then, we generate 20,000 labeled sentences by filling these templates with the groundtruth attribute label. On the augmented user comment data, we train the user simulator.

Our user simulator is implemented via a sequence-to-sequence model. The inputs of the user simulator are the differences on one attribute value between the candidate and desired items. Given the inputs, the user simulator generates a sentence describing the visual attribute difference between the candidate item and the desired item. We use two LSTMs as the encoder and decoder. The dimensionality of the latent code is set as 256. We use Adam as the optimizer, where the initial learning is set as 0.001 with batch size of 64. Note that for evaluating how the current recommended item's visual attributes satisfy the user's previous feedback, our user simulator on UT-Zappos50K only generates simple comments on the visual attribute difference between the candidate image and the desired image: we can calculate how many attributes violate the users' previous feedback based on the visual attribute groundtruth available in UT-Zappos50K.

We define four evaluation metrics: *i*) task success rate (SR@$K$), which is the success rate after after $K$ interactions; *ii*) number of user interactions before success (NI); and *iii*) number of violated attributes (NV). In each user session, we assume the user aims to find items with a set of desired attribute values sampled from the dataset. We report results averaged over 100 sessions with standard error. We develop an RL baseline approach by ignoring the constraints (*i.e.*, discriminator) in RCR. A major difference between our RL baseline approach and Guo, *et al.* [19] is that we consider the attributes in the model learning, while the attributes are ignored in [19]. We compare RCR with the RL without constraints, as well as RL methods with naive constraints, *i.e.*, naively using hard constraints. That is, we track all the visual attributes previously described by the user in this session, and make further recommendations based on the matching between them and the items in dataset.

**Analysis** All models are trained for 100,000 iterations (user sessions), and the results with standard errors under different metrics are shown in Table 1. The proposed RCR framework shows consistent improvements on most metrics, compared with the baselines. The gap between RL with naive constraints and RCR demonstrate the learned constraint (discriminator) has better generalization. Figure 4 shows the metrics with standard errors of RL and the proposed RCR in the first 40,000 iterations. RCR shows much faster convergence than RL. The last subfigure shows the values of $\lambda$ with different number of samples. It is interesting to see that $\lambda$ increases at the initial stage because of too many violations. Then, with less violations, $\lambda$ decreases to a relatively small value as

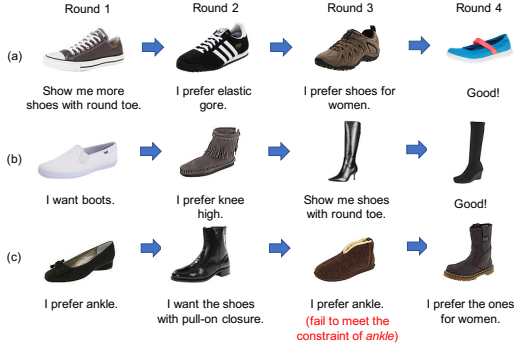

Figure 5: Three use cases, from logged experimental results. (a) and (b) are successful use cases by RCR. (c) is not successful by RL, which demonstrate the common challenge of failing to meet the constraint in recommendation.

$\lambda = 0.04$ and then remains stable, which is the automatically learned weight of the discriminator. Some examples in Figure 5 show how the constraint improves the recommendation.

## 5.2 Constrained Text Generation

**Experimental Setup** We use the Yelp review dataset [40] to validate the proposed methods. We split the data as 444,000, 63,500, and 127,000 sentences in the training, validation and test sets, respectively. The generator is trained on the Yelp dataset to generate reviews *without sentiment labels*. We define the reward of the generated sentence as the probability of being real and the constraint is to generate positive reviews, *i.e.*, the generator will receive a penalty if it generates negative reviews. The constraint is a neural network with a classification accuracy of 97.4% on the validation set, trained on sentences with the sentiment labels. We follow the strategy in [52, 18] and adopt the BLEU score, referenced by test set with only positive reviews (test-BLEU) and themselves (self-BLEU) to evaluate the quality of generated samples. We also report the violation rate (VR), the percentage of generated negative reviews violating the constraint.

| | Test-BLEU-2 | 3 | 4 | 5 | Self-BLEU-2 | 3 | 4 | VR |
|---|---|---|---|---|---|---|---|---|
| RL | 0.807 | 0.622 | 0.469 | 0.376 | 0.658 | 0.315 | 0.098 | 40.36% |
| RCR (ours) | 0.840 | 0.651 | 0.492 | 0.392 | 0.683 | 0.348 | 0.151 | 10.49% |

Table 2: Comparison between RCR and standard RL for constrained text generation on Yelp.

**Analysis** As illustrated in Table 2, RCR achieves better test-BLEU scores than standard RL, demonstrating high-quality generated sentences. Further, RCR shows a little higher but reasonable self-BLEU scores, since we only generate sentences with positive sentiments, leading to lower diversity. Our proposed method shows much lower violation rate, demonstrating the effectiveness of RCR. Some randomly generated examples are shown in Table 3.

| RL without Constraints | RCR |
|---|---|
| *the ceiling is low , the place smells awful , gambling sucked .* | every dish was so absolutely delicious and seasoned perfectly . |
| i have been here a few times and each time has been great ! | he is the most compassionate vet i have ever met . |
| *bad food , bad service , takes too much time .* | compared to other us cities , this place ranks very generous in my book . |
| *food was good , but overall it was a very bad dining experience .* | then you already know what this tastes like . |
| my entree was a sea bass which was well prepared and tasty . | thank you my friends for letting us know this finest dining place in lv . |
| the food is delicious and very consistently so . | great service and the food was excellent . |
| *the waitress was horrible and came by maybe once every hour .* | the lines can get out of hand sometimes but it goes pretty quick . |

Table 3: Randomly selected examples of text generation by two methods.

## 6 Conclusions

Motivated by potential constraints in real-world tasks with RL training, and inspired by constrained policy optimization, we propose the RCR framework, where a neural network is parameterized and dynamically updated to represent constraints for RL training. By applying this new framework to constrained interactive recommendation and text generation, we demonstrate that our proposed model outperforms several baselines. The proposed method is a general framework, and can be extended to other applications, such as vision-and-dialog navigation [44]. Future work also includes incorporating user historical information into the recommendation.

## Footnotes

[2] For simplicity, we here only introduce one constraint function; in practice, there may be many constraint functions.

[3] Available at `https://github.com/hongwang600/image_tag_dataset/tree/master/tags`.

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
