[Supplementary Material · NIPS2019CameraReady-4_supp.pdf]

# A  More details of the RCR Model

## A.1  Constrained Text Generation Model

A content vector $z_x$ is given by an encoder $\text{Enc}(\cdot)$, with inputs $X \sim \mathcal{D}$, *i.e.*, $z_x = \text{Enc}(X)$. Based on $z_x$, an LSTM-based [23] decoder $G(\cdot)$ generates a new sentence $\tilde{Y}$ that is expected to be the same as the inputs. The auto-encoder can be trained by minimizing the following reconstruction loss:

$$\mathcal{L}_{ae}(\boldsymbol{\theta}) = \mathbb{E}_{X \sim \mathcal{D}}[-\log p_{\boldsymbol{\theta}}(X|z_x)]. \tag{12}$$

**Constraint Discriminator: Style Classifier**  The constraint discriminator is trained to detect the violation of the constraints and we consider a classifier $C_\phi$ as the discriminator. The classifier is trained to distinguish sentences with violation and non-violation set $p_r$ and $p_f$ respectively:

$$L(\phi) = -\mathbb{E}_{(X) \sim p_f} \left[ \log(C_\phi(X)) \right] - \mathbb{E}_{(X) \sim p_r} \left[ \log(1 - C_\phi(X)) \right]. \tag{13}$$

**General Discriminator: Language Model**  Language model can work as a general discriminator [48], which learns a conditional distribution over the current word given previous words in a sequence. Let $p_l(\hat{Y})$ be the probability of a sentence $\hat{Y}$ evaluated with the language model $D_l$. We have

$$\mathcal{L}_{lm}(\boldsymbol{\theta}) = \mathbb{E}_{X \sim \mathcal{D}, \hat{Y} \sim p_{\boldsymbol{\theta}}(z_x)}[-\log p_l(\hat{Y})]. \tag{14}$$

## A.2  Constrained Interactive Recommendation

**Constraint Discriminator**  We have several ways to build up the violation and non-violation sets for the training of the constraint discriminators in interactive recommendation. We can collect the datasets from two distributions as described in Assumption 1.

**Assumption 1** *If a user is not satisfied with current recommendations based on her natural language feedback, then the current recommendation violates the user preference. Further, all desired items do not violate the corresponding user historical feedback.*

Besides, we can exploit huge-amount offline data, which is available in certain cases in real-world. Based on the attributes information, we can build up these two datasets. The performance of different ways to collect data and train the discriminator is similar in our case.

**User Simulator**  We derive the templates from the real-world sentences collected from annotators. Some examples of the templates are

- Please show me more ______.
- I am looking for ______.
- I prefer ______.
- I want the shoes with ______ closure.
- ...

The visual attributes used in our evaluation include shoes category, shoes subcategory, heel height, closure, gender, and toe style. There are 4, 21, 7, 18, 8, 19 classes for these attributes, respectively.

# B  Ablation Study

We perform an ablation study to understand how $\lambda_{\max}$ affects the performance. To remove the affects by the discriminator in this ablation study, we experiment different values of $\lambda_{\max}$ without using the discriminator to reject the items sampled from the recommender. The results are shown in Figure 6. When $\lambda_{\max} \leq 0.05$, increasing $\lambda_{\max}$ leads to improved performance under NI, NV, SR. When $\lambda_{\max} > 0.05$, only minor improvements can be observed by increasing $\lambda_{\max}$. This matches the observation in Figure 4 that the value of $\lambda$ fluctuates around $0.04$ after $\lambda$ being updated on about $1,800$ samples. Besides, we observe that when $\lambda_{\max} = 0.01$, the training is much more computationally expensive since the agent can not succeed and terminate the episode earlier in most cases. This experiment validates that our algorithm can adaptively find a suitable $\lambda$ to balance the weight between the reward and constraint, which leads to more efficient model learning.

Figure 6: Number of Interactions (NI), Number of Violations (NV), Success Rate (SR) with respect to training iterations with different $\lambda_{\max}$.

## C Generated Examples

We show some examples of the generated feedback by the user simulator in Figure 7 and Table 4. To evaluating how the recommended item's visual attributes satisfy the user's previous feedback, our simulator only generates simple comments on the visual attribute difference between the candidate image and the desired image: we can calculate how many attributes violate the users' previous feedback based on the visual attribute groundtruth available in UT-Zappos50K.

Show me more shoes with laced up closure.

I prefer shoes with round toe.

I am looking for ankle.

I want the shoes with ankle strap.

I prefer 3in - 3 3/4in.

Show me more shoes with close toe.

Figure 7: Examples of the generated feedback by the user simulator.

| Round | Simulated User Feedback | Round | Simulated User Feedback |
|---|---|---|---|
| 1 | I am looking for shoes for women. | 1 | I am looking for shoes for men. |
| 2 | I prefer heels. | 2 | I am looking for shoes with lace up. |
| 3 | Please show me more shoes with high heel. | 3 | Do you have shoes with medallion. |
| 4 | I want the shoes with slip-on closure. | 4 | I am looking for shoes with flat. |
| 5 | I prefer high heel. | 5 | Do you have flat. |
| 6 | I prefer shoes with pointed toe. | 6 | Show me more shoes with flat. |
| 1 | Do you have shoes with open toe. | 1 | Do you have shoes with shoes. |
| 2 | Please provide some shoes for girls. | 2 | I prefer round toe. |
| 3 | Do you have more shoes for girls. | 3 | Do you have clogs and mules. |
| 4 | I want hook and loop. | 4 | Show me more shoes with slip-on. |
| 5 | - | 5 | Show me more shoes with slip-on. |
| 6 | - | 6 | - |
| 1 | Do you have shoes with open toe. | 1 | I want sneakers and athletic shoes. |
| 2 | Please provide some shoes for girls. | 2 | Do you have shoes with lace up. |
| 3 | I am looking for shoes for girls. | 3 | Do you have shoes for men. |
| 4 | Do you have more shoes for girls. | 4 | Do you have shoes with center seam. |
| 5 | Please provide some shoes with center seam. | 5 | Show me more shoes with center seam. |
| 6 | - | 6 | I like 1in - 1 3/4in. |
| 7 | - | 7 | Show me more shoes with 1in - 1 3/4in. |
| 8 | - | 8 | I am looking for shoes with knee high. |
| 9 | - | 9 | Do you have more shoes with knee high. |

Table 4: Examples of the generated feedback by the user simulator.