[Reviews · NeurIPS 2019]

Reviewer 1



The paper is well described and the proposed method appears to be interesting and useful. The authors show performance improvement over existing methods on text-based recommendation and text generation tasks. I'd suggest more details to be provided for "Model Training" (line 125-151), including explaining the selection of \Gamma and \lambda_{max} and how the projection operator stabilizes the parameters.

Reviewer 2



This paper presents a novel reinforcement learning algorithm that incorporates constraints from previous feedback into recommendation. The key contribution is that the user preference constraints can be learned by a discriminator neural network. It is well written and easy to understand, and experimental results clearly demonstrate that the proposed method outperforms other existing baseline algorithms.

Reviewer 3



The paper proposes a new Constraint Reinforcement Learning method, where the constraints are sequentially added. The authors apply this method in the text-based recommendation and text generation tasks. Generally speaking, this paper is well-written and well organized. The motivation example (i.e. feedback such as clicking or rating contains little information to reflect complex user attitude towards various aspects of an item) in Introduction naturally motivates the proposition of the sequentially added constraints. The proposed model (i.e. Eq. (3), Eq. (5) and its model details) is consistent with the target task. The reward and constraints are reasonably designed. The experimental setting is remarkable (especially the Online Evaluation by simulator and the four proposed evaluation metrics) and the results are positive. However, this paper still has the following minor issues. 1. In the experiment section, the sentences are generated by a GRU model. Then how to ensure that the sentences are short and meaningful such as ‘ankle boots’ and ‘shoes with suede texture’ in Fig. 2. 2. It seems that the recommendation completely depends on the current visual picture and natural language feedback, without considering historical behaviors as in traditional recommendation methods. I wonder whether it can outperform traditional recommendation methods, or how to incorporate them into the framework in this paper. 3. More related works need to be cited. ‘Q&R: A two-stage approach toward interactive recommendation, KDD 2018’ and ‘Query-based Interactive Recommendation by Meta-Path and Adapted Attention-GRU, arxiv 2019’ also focus on interactive recommendation. Constraints are added in a sequential fashion and the recommendation is based on the sequential constraints. Therefore, sequential recommendation such as ‘What to Do Next: Modeling User Behaviors by Time-LSTM, IJCAI 2017’ and ‘A Brand-level Ranking System with the Customized Attention-GRU Model, IJCAI 2018’ are also related to this paper. 4. Typos: a. Page 2: ‘In text-based recommendation ,’ should be ‘In text-based recommendation,’; ‘, in practice’ should be ‘. In practice’. b. Page 3: ‘proposed to using’ should be ‘proposed to use’. c. Page 5: ‘p(X)’ should be ‘p(\hat{X})’. d. Page 6: ‘ Besides’ should be ‘. Besides’; ‘manual’ should be ‘manually’; ‘improving’ should be ‘improve’; ‘ahierarchical’ should be ‘a hierarchical’; ‘softmaxand’ should be ‘softmax and ’. e. Page 7: ‘after after’ should be ‘after’. After the authors' response: Most of my questions are well answered except for the following two. 1. Although the authors argue that the GRU model is trained on short sentences and would generate short sentences, in the format of simple sentences with prefix. I still doubt whether it can surely generate meaningful short sentences. 2. I do not agree that 'Traditional recommendation models are usually trained in an offline manner'. Because there are some recommendation methods that are trained in an online learning fashion, and meanwhile, they consider historical behaviors, e.g. FTRL. However, these two minor issues do not prevent this paper from being a good submission. I would not change my score.

[Author Response · NeurIPS 2019]

We thank all the reviewers for their positive comments, and address their major questions and comments below. Clarifications will be added in the revision and we will keep improving our draft.

**Reviewer #1** We thank the reviewer for the positive reviews. The remarks raised are addressed below.

**Q: More details about model training**

**A:** $\lambda_{\max}$ is selected via parameter search for $\lambda_{\max} \in \{0.1,\ 0.5,\ 1.0,\ 2.0,\ 5.0\}$, ending up with $\lambda_{\max} = 1.0$. For $\Gamma_\theta(\cdot)$, we use PPO in our implementation to keep the stability as the parameters are updated within a 'trust region'.

We are happy to release our code for better reproducibility.

**Reviewer #2** We appreciate reviewer's acknowledgement of our novelty and suggestions provided.

Thanks for the suggestion on the paper improvements. In our updated version, we will add more results to show how our approach can handle the more useful dialogue generation problem.

**Reviewer #3** We thank the reviewer for the positive reviews and appreciate the reviewer's suggestions.

**Q: How to ensure that the sentences are short and meaningful.**

**A:** Our paper proposed a general constraint-augmented reinforcement learning framework. In the recommendation task, natural language is only an example way for the user to express the preferences (*i.e.*, constraints), and we do not focus too much on how to handle more complicated or even free-form language. Therefore, similar to [20], our sentences in experiments are in the format of simple sentences with prefix. In this setting, the length of the sentences is implicitly determined by the training data. That is, we train the GRU model (the user simulator) on short sentences collected by human, and thus the trained model usually generates short sentences.

**Q: Whether historical behaviors are considered.**

**A:** Our approach considers the user historical behaviours within the current user session, although it does not model the user historical behaviours in previous sessions. The GRU tracks the user behaviours and the discriminator considers all previous user preferences in the current user session. However, we do agree that user historical behaviours from previous sessions can be employed to further enhance the performance.

**Q: Comparison with traditional methods considering historical behaviors.**

**A:** Our approach is not directly comparable to the traditional methods considering historical behaviors. Traditional recommendation models are usually trained in an offline manner, *i.e.*, the model is trained on a pre-collected dataset. By contrast, our method is proposed in a different setting, where the pre-collected dataset is not required and our recommender is interactively learned when the user interacts with the system. Moreover, it is not clear how to handle the interactive natural language feedback to provide interactive recommendations by traditional models.

**Q: Incorporating traditional methods into our framework.**

**A:** A simple approach can be developed to incorporate the traditional recommendation methods into our framework, to leverage historical behaviours from previous sessions. Assume we have users' historical behavior data and train a traditional model. In each user session, we make initial recommendations by the traditional model, collect natural-language feedback from users, and make further interactive recommendations by our framework. After a number of user sessions, we can update the traditional model based on the recently collected users' feedback to the items.

Thank you very much for providing us with the related work [49, 50, 51, 52]. We will definitely discuss these references in our related work section, and fix all the typos in our minor revision.

[49] Konstantina Christakopoulou, Alex Beutel, Rui Li, Sagar Jain, and Ed H Chi. Q&R: A two-stage approach toward interactive recommendation. In KDD, 2018.
[50] Yu Zhu, Yu Gong, Qingwen Liu, Yingcai Ma, Wenwu Ou, Junxiong Zhu, Beidou Wang, Ziyu Guan, and Deng Cai. Query-based interactive recommendation by meta-path and adapted attention-gru.arXiv:1907.01639, 2019.
[51] Yu Zhu, Hao Li, Yikang Liao, Beidou Wang, Ziyu Guan, Haifeng Liu, and Deng Cai. What to do next: modeling user behaviors by time-lstm. In IJCAI, 2017.
[52] Yu Zhu, Junxiong Zhu, Jie Hou, Yongliang Li, Beidou Wang, Ziyu Guan, and Deng Cai. A brand-level ranking system with the customized attention-gru model. In IJCAI, 2018.


[Meta-Review · NeurIPS 2019]

The paper describes a constraint augmented RL technique for text generation. This is excellent work. The paper is well written. The algorithm is novel and significant. The experiments are convincing. Well done!